# Mobile X-ray Outside the Hospital vs. X-ray at the Hospital Challenges Exposed in an Explorative RCT Study

**DOI:** 10.3390/healthcare8020118

**Published:** 2020-04-30

**Authors:** Maria Toppenberg, Thomas Christiansen, Finn Rasmussen, Camilla Nielsen, Else Marie Damsgaard

**Affiliations:** 1The Department of Radiology, Aarhus University Hospital, 8200 Aarhus, Denmark; thomchri@rm.dk (T.C.); firasm@rm.dk (F.R.); 2DEFACTUM, Social and Health Services and Labour Market, 8200 Aarhus, Denmark; camilla.palmhoj@rm.dk; 3Department of Geriatrics, Aarhus University Hospital, 8200 Aarhus, Denmark; elsedams@rm.dk

**Keywords:** mobile X-ray, nursing homes, frail elderly, RCT, healthcare

## Abstract

Background: For frail patients, it may sometimes be preferable to carry out X-ray examinations at the patients’ own home. The general state of such patients may worsen due to transport and change of environment when transported for examination at the hospital. Objective: The aim of the randomized controlled trial (RCT) was to investigate if mobile X-ray improves healthcare for fragile patients. The primary outcome was the number of hospitalizations. Data sources: We collected all data using questionnaires and data from the Electronic Patient Record (ER). Participants: Patients referred to a mobile X-ray examination living in nursing homes and homes for the elderly in the Aarhus Municipality (Denmark). Intervention: mobile X-ray examinations compared to those at the hospital. Study appraisal: Data were collected and stored using the computer programme Redcap. Stata was used for statistical calculations. One hundred and thirty-six patients were included in the RCT. We did not find significant differences between mobile X-ray (intervention) and X-ray at the hospital (control) concerning hospitalizations and number of hospital days. Challenges: We met several challenges when carrying out RCT in the planned study population. Doctors often withdraw the referral when they found out that their patient should go to the hospital instead of mobile X-ray. The nursing home staff often considered the patient too frail to allow the test person to ask questions post X-ray. We also met challenges in the randomization method resulting in bias in the first data collection, so we had to adjust the randomization method. Conclusions: For the fragile patients in the present explorative study, mobile X-ray did not significantly reduce the number of hospitalizations compared to X-ray at the hospital. Yet, mobile X-ray may be a new important diagnostic tool for more precise treatment to the frailest patients for whom transportation to the hospital is too exhausting. We need studies with focus on this aspect. We also recommend future RCT studies in a population for which mobile X-ray has not yet been a possibility.

## 1. Introduction

It sometimes may be preferable for fragile patients to have the X-ray examination carried out at their own home. Fragile patients may worsen due to the transportation and environmental change related to examination at the hospital [1,2]. This might lead to increased need for care and medication for several days after the examination at the hospital. Furthermore, the primary disease may deteriorate, and delirium develops. Even worse, some patients will need extended time of hospitalization [2,3,4]. In the Western World, mobile X-ray is offered to the frail elderly, homeless, drug users, asylum seekers, and nursing home residents outside the hospital. Patients and health care staff may experience benefits when using mobile X-ray. Benefits are reported in both qualitative and quantitative studies such as decreased number of patients being hospitalized, decreased number of patients becoming delirious and increased number of examined patients [2,5,6,7,8,9,10,11,12,13,14,15,16,17]. As mobile X-ray is performed using transportable equipment, the quality is not as good as using standard equipment at the hospital, which is the gold standard. Though studies report that radiologists are able to diagnose patients reading images performed with mobile X-ray equipment and making their use technically feasible. None of these studies were randomized controlled trials comparing mobile X-ray and traditional X-ray at the hospital including both skeletal and thorax images [8,11,18,19,20]. One quasi randomized study of mobile X-ray was published in 2006, but only with focus on nursing home patients having thorax X-rays examinations. Furthermore, a cross-sectional study found that diagnosing nursing home residents for pneumonia, mobile X-ray was feasible, but clinicians still need to trust the clinical signs before ordering a chest X-ray [8,11,18,19,20].

At the department of radiology at Aarhus University Hospital, mobile X-ray has been offered since 2014. The patients were examined in their own home using mobile X-ray. The images were sent to the department of radiology where radiologists would read and make quality control before the mobile unit left the patient. No images using mobile X-ray needed to be repeated.

The target groups were frail elderly patients living in nursing homes, patients living in homes for the elderly, homeless patients living in care homes, handicapped patients living in homes for handicapped, psychiatric patients hospitalized in the psychiatric ward and patients in a ward for patients with a need of rehabilitation after acute stroke and hip fracture placed at a location 3 km from the main hospital. After one year the project was considered valuable to be continued because mobile X-ray seemed to improve the healthcare for fragile patients, but the conclusion was that the effect should be tested in a randomized controlled trial (RCT) [21]. Since the literature lacks the evidence for documenting the effect of mobile X-ray, we wanted to do a RCT of mobile X-ray of frail elderly, both thorax and skeleton [22].

### Aim

The aim of the RCT was to investigate if mobile X-ray at home improved healthcare defined by hospital bed days as the primary outcome for fragile patients compared to X-ray at the hospital. We also wanted to describe the challenges connected to use RCT to measure the effect of mobile X-ray of the fragile elderlies.

## 2. Methods

### 2.1. Study Design

Our study was a non-blinded RCT. Patients were included from The University Hospital in Aarhus.

### 2.2. Study Population Included

The study population included patients aged 65+ years.
Referred to a mobile X-ray examination.Living in nursing homes and homes for the elderly in Aarhus municipality.

Exclusion criteria: Patients already hospitalized in the University Hospital, homeless patients, handicapped patients and patients already examined with mobile X-ray in the study period.

The study was approved by the Ethical Committee (53 811) and Data protection system (1-16-02-124-15) in the Central Denmark Region and registered in Clinicaltrials.gov (NCT04005040). Involved Departments at Aarhus University Hospital gave their consent for looking in the Electronic Patient Journal (EPJ).

### 2.3. Outcome Measures

The outcome data were collected by a data collector, trained for the purpose. The outcome measures were validated for use primarily in the elderly population, but also tested in the specific patient group before the study began. All outcome measures were tested for normality. The following outcomes are reported in the article:

#### 2.3.1. Baseline

Age, gender, number of diagnosis, polypharmacy ≥5, patients examined by doctor before X-ray, diagnosed with dementia, BMI ≤ 21, MMSE, GDS, MBI and Cirs G.

#### 2.3.2. Primary Outcome

Hospital admission (yes/no): data were collected from local systems at the hospital [17].

Number of hospital days (date of admission until date of discharge): data were collected from local systems of the hospital [17].

#### 2.3.3. Secondary Outcomes


Confusion Assessment Method (CAM) for detecting delirium [23].Depression List (DL) designed to measure quality of life among patients with a MMSE score down to 5 and data is collected by proxy [24,25,26].Completion of examination (yes/no).Death (yes/no) (within one week after X-ray).


### 2.4. Randomization and Blinding

Patients were randomized to either mobile X-ray in their own home or to X-ray at the department of radiology of The University Hospital. The data collection was divided into two parts.

Part one: Patients were randomized by days. One day all patients were randomized to mobile X-ray and another day all patients were randomized to X-ray at the hospital. Patients and data collected in part one were not included in this study because this method resulted in bias, because unfortunately general practitioners (GP) were able to influence patients assignment to the study. After approval from the Danish Ethical Committee we changed the method.

Part two: In the second data collection the computer programme Redcap (a safe system to build up databases in science) was used to randomize patients by a computer-generated block randomization with blocks of 8 in each group. Using this method, GPs could not influence patients’ assignment to the study. It was not possible to blind everybody in the study.

#### 2.4.1. Intervention

The intervention group comprised patients examined using a mobile X-ray in their own home.

#### 2.4.2. Control

The control group comprised patients examined using X-rays at the hospital.

### 2.5. Statistical Analysis

The literature to guide the power calculation was limited. The only randomized controlled study with portable X-ray in year 2006 was the study by Loeb [17]. Its limitation was that the study population was suspected only of pneumonia and not a broad pathology [17]. Based on the study by Loeb the expectation was that 10% of patients examined with mobile X-ray and 22% of the patients examined with X-ray at the hospital were hospitalized. To show such a difference at a significance level of 5% (two-sided test) with a power of 80% would require a sample size of 374 patients (187 patients in each group) [27,28]. Unfortunately, we did not reach the required sample size within the time limit, because we had to start the data collection twice. The sample size calculation and statistical analysis were done using the statistical software package Stata version 13.1 supervised by a statistician (StataCorp LP, College Station, TX, USA).

Continuous baseline variables were analysed with the two-sample t-test. Categorical post examination characteristic variables were analysed by the chi2 test, with the intervention group as the explanatory variable. The distributions of the primary independent outcome variables were not normal and therefore the Wilcoxon’s Rank Sum test was used. A two-sided *p*-value below 0.05 was set as the indicator of statistical significance. The intervention group was compared to the control group without match-pairing of the X-rays.

The dataset contains missing values and might bias the results. To increase the effect of the statistical tests, we analysed if the values according to the worst case scenario in the two study populations would change the result in the effect variables using a sensitivity test [27,28]. In the mobile X-ray group we had the following missing data: CAM (n = 13), DL (n = 28), MMSE (n = 29), GDS (n = 29), MBI (n = 11), CIRS G (n = 11). In the X-ray at hospital group we noted the following missing data: CAM (n = 23), DL (n = 33), MMSE (n = 36), GDS (n = 36), MBI (n = 11), CIRS G (n = 11).

### 2.6. Inclusion of Participants

The inclusion period started the first of May 2018 and lasted until 12 April 2019. In that period 177 records were randomized to be included in the study. In Figure 1 the patient inclusion flow is shown.

We had challenges including patients for several reasons. For instance when doctors found out that their patient was randomized to X-ray at the hospital, some of the doctors withdrew the referral or called the Department of Radiology to make them change the X-ray method to mobile X-ray. Therefore, we conducted 10 semi-structured interviews with randomly selected GPs who had referred a patient to mobile X-ray and the patient was randomized to X-ray at the hospital. We used an interview guide with the same five questions to identify barriers for referring patients. The five questions were:Are you satisfied with mobile X-ray?Do you know the quality of mobile X-ray?Do you understand why we have to randomize patients in the study of mobile X-ray?Why would you withdraw the referral of a patient?Could a patient be too fragile to be examined at the hospital?

### 2.7. Ethics Approval and Consent to Participate

The study was approved by the Ethical Committee (53 811) and Data protection system (1-16-02-124-15) in the Central Region and registered in Clinicaltrials.gov (NCT04005040).

## 3. Results

### 3.1. Baseline Characteristics

Baseline characteristics collected post X-ray examination within one week after X-ray examination (Table 1). There were no statistically significant differences between the two groups. Data were obtained in the patients’ own home by interviewing healthcare staff and directly registered in RedCap, a computer programme also recommended for collecting data. To gain completeness of baseline data, these were supplemented with data from local data systems.

Table 2 shows mean score for Mini Mental State Examination (MMSE) [26], Geriatric Depression Scale (GDS) [29], Modified Barthel Index (MBI) [30] and Cumulative Illness Rating Scale-Geriatric (Cirs G) [31]. Table 2 also show test scale and impute values for the sensitivity analysis. All data were collected after X-ray examination and therefore MMSE and GDS could be biased because the patient might have been affected by the examination.

Mini mental state examination (MMSE) [26] and Geriatric Depression Scale (GDS) [25,32] were obtained by testing the patient. Statistical difference was found between the intervention group and control groups concerning GDS, but not for MMSE. The sensitivity analysis did not change the baseline characteristics between the two groups in MMSE to be statistically significant (*p* = 0.32). Modified Barthel Index (MBI) [30] and Cumulative Illness Rating Scale (CIRS) [31] were obtained by interviewing the healthcare staff. The sensitivity analysis did not change the baseline characteristics between the two groups for MBI to be statistically significant (*p* = 0.11) or CIRS (*p* = 0.77) by using 0.05 as the standard for significance. Neither of these was statistically significant.

### 3.2. Primary Outcome

We did not find statistically significant differences between the two groups within one week after the X-ray examination (*p* = 1.00) (Table 3). The reasons why the patients in the intervention group were hospitalized were: aplastic anaemia, dyspnoea (severe pneumonia), fracture in femur, fractures in the ribs (hospitalization was due to a fall and the patient complained of much pain elsewhere), bacterial pneumonia, pleura effusion (two patients) and hypoglycaemia. The reasons why the patients in the control group were hospitalized were: fracture in femur (three patients), abscess in thigh, erysipelas, cystitis, flue with virus A and abnormal findings in the abdomen/retroperitoneum.

In Table 4 no statistically significant difference was observed between the two groups of patients concerning number of hospital days (*p* = 0.12).

### 3.3. Secondary Outcomes

Table 5 shows that we did not find significant differences between the two groups concerning the secondary outcomes. Sensitivity analysis for CAM and DL including all randomized patients and assuming worst possible measured outcome changed the result of CAM to be statistically significant (*p* = 0.023), but this was the opposite for DL (*p* = 0.24), for which the result did not become statistically significant.

### 3.4. Challenges

One of the challenges was a balanced recruiting of patients to the two groups. Mobile X-ray had become a standard offer prior to initiating the RCT. The referring doctors understood the meaning and importance of the RCT of mobile X-ray. But in the interviews the doctors said that the uncertainty of referring a patient to mobile X-ray without knowing if the patient would be examined with mobile X-ray was an uncertain situation for the patient, health care staff and relatives. Some patients were too weak to be transported to the hospital. If the patient was very weak and randomized to the hospital group, the doctor often had chosen to start treatment without X-ray examination.

Another challenge was the nursing home staff wishing to protect the frailest patients. In some cases, the data collector was not permitted to see and test the patients. In some cases, the staff were too busy to participate in the interview leading to missing data.

## 4. Discussion

This RCT of mobile X-ray of both lungs and skeletal in nursing home residents is the first conducted to our knowledge. We did not find significant differences in our primary outcome hospitalizations, which are in contrast to Loeb et al., who conducted a cluster-randomized controlled study. The observed prevalence of hospitalization in our intervention group (11.76%) can be compared to Loeb et al., where 10% in the mobile X-ray group was hospitalized. In their control group 22% was hospitalized compared to 12% in our control group [17]. A reason why the results are skewed could be that those already at the hospital were easier to keep there than those outside the hospital, where the easier decision is just to treat in place. The lower percentage in our control group may also be explained by the fact that the most frail patients in our study were treated by the doctor in their nursing homes without X-ray and without a higher death rate. Another reason could be that we included both thoraxes (31%) and skeletal examinations (69%) while Loeb et al. only included thorax examinations. We found an insignificant tendency toward a longer mean hospital stay in patients examined at the hospital (7.63) compared to patients examined with mobile X-ray (4.50). Our result may be explained by the possible under powering of our study. Loeb et al. included 680 patients referred from nursing homes suspected for pneumonia compared to our 134 patients of whom one third were referred due to suspicion of pneumonia [17]. On the other hand, maybe a cluster-randomized study was a better design for studying mobile X-ray in nursing home residents.

The non-significant difference in delirium measured by CAM and the fact that only one patient in the control group was classified as delirious after X-ray examination is in contrast to the result by Ricauda et al. In that pilot study of home X-ray of thorax they found a significantly higher risk of delirium among patients examined with X-ray at the hospital [13]. In the sensitivity analysis we found significant results in CAM score (*p* = 0.023) so our result might be explained by the large number of missing values which seems to be due to staff protecting delirious patients against the post X-ray tests by the data collector. The result might also be explained by the fact that the two study populations were not comparable concerning home, age, GDS score and CIRS score. Our study population was living in nursing homes and not at their own home. Our patients were older, less depressed, had higher pre morbidity, but the patients had the same mean MMSE score in the study by Ricauda (22.5) [13]. In Table 1 baseline characteristics show that 40% in the intervention group and 45% in the control group was diagnosed with dementia. This result is comparable to Table 2 where 57% in the intervention group and 45% in the control group was tested. This fact supports the theory of staff protecting fragile patients but also resulted in missing values and may bias the data.

We found a higher quality of life measured by DL in the mobile X-ray group in spite of the missing values (*p* = 0.05), but this difference disappeared in the sensitivity analysis taking missing data into account. This still need to be examined in further studies.

### 4.1. Challenges and Future Studies

This RCT study share some of the same challenges as other primary research studies conducted in a frail and weak patient group. Specific issues encountered here are related to the quality and validity of data.

### 4.2. Study Population and Outcome Measures

Previous researchers have highlighted the challenges of defining the study population and the outcome measures [8,11,13,16]. Our intention was to include all subgroup patients referred to mobile X-ray in Aarhus municipality. We found out that it was challenging, because one outcome measure could be significant in one group and it might not even be relevant in another group, as the patients are so different, so future studies may need to include several outcomes before choosing a primary outcome for the specific study population. The aim of our RCT was to investigate if mobile X-ray improves healthcare. Hospitalization was chosen as the primary outcome because according to the literature, this was the most qualified measurable outcome to study if healthcare was improved [17]. No other RCT studies are directly comparable to our study, but Loeb et al. conducted a cluster-randomized study of mobile X-ray using hospitalizations as outcome measure. Our control group could be compared to their control group concerning age, residence and gender. Our intervention group could be compared to their intervention group concerning residence and gender, but our intervention group was younger. Our RCT mobile X-ray did not result in significant reduction in the number of hospitalizations compared to X-ray at the hospital. Based on this study we are not sure that hospitalization is the most suitable outcome. Other outcome measures for instance quality of life, satisfaction or economic perspectives could be relevant in future studies of mobile X-ray. Furthermore, it should be studied if mobile X-ray gives the doctor an extra diagnostic help leading to a more precise treatment in the frailest patients being too ill for transportation to the hospital. We also recommend that future studies involve patients when choosing outcomes.

A limitation of the study is that the quality of the radiogram was not evaluated with a comparison between the two modalities mobile X-ray and X-ray at the hospital. The reason why was limited time and money, but the study is possible and ready to conduct.

Our results show that the effect of mobile X-ray depends of the study population and the participants’ ability to participate. However, for evaluating mobile X-ray another study design could be considered for instance a cluster randomized study as Loeb et al. [17].

### 4.3. Referring Doctors

High intervention implementation turned out to be a challenge, because the project depended on doctors referring patients without withdrawal of patients randomized to the hospital group. The study group might be skewed against fewer patients in the hospital group including severe illness and reduced cognitive functions. We saw a tendency among doctors to withdraw the referrals for patients randomized to the hospital group in the first data collection period. Future evaluations of mobile X-ray might have to be carried out in a setting where mobile X-ray has not been offered before, so no one expect a certain examination method. We recommend this because we had so many challenges recruiting patients.

### 4.4. Missing Values

It was both a strength and a limitation that we were allowed by the Danish Ethical Committee to conduct the RCT without patient consent. It was a strength because getting informed consent from frail cognitive impaired patients in time before an acute examination is almost impossible. It turned out to be a limitation because the healthcare staff of nursing homes often said that the patient was too weak to participate in the test after X-ray examination which lead to missing data.

Assuming that the patients were weakened and ill we had healthcare staff participating on behalf of the patients in order to improve the quality of data. Still we had a large number of missing data, probably resulting in fewer statistically significant results. In future studies, baseline characteristics may have to be collected using national registers instead of asking frail patients or/and busy healthcare staff.

## 5. Conclusions

Our RCT did not show significant differences in hospitalizations between the intervention group and the control group for nursing home residents in Aarhus Municipality in Denmark. Our results should only be regarded as exploratory and subject for further research of mobile X-ray. The results of our study showed that a RCT of a mobile X-ray already being settled as a diagnostic offer has several challenges. On the other hand, mobile X-ray may help doctors to increase the diagnostic safety in frail older patients for whom transportation to hospital is so big a challenge that it will not be done. This should be focused on in future studies. Furthermore, choosing the right study design is more important rather than conducting a RCT.

## Figures and Tables

**Figure 1 healthcare-08-00118-f001:**
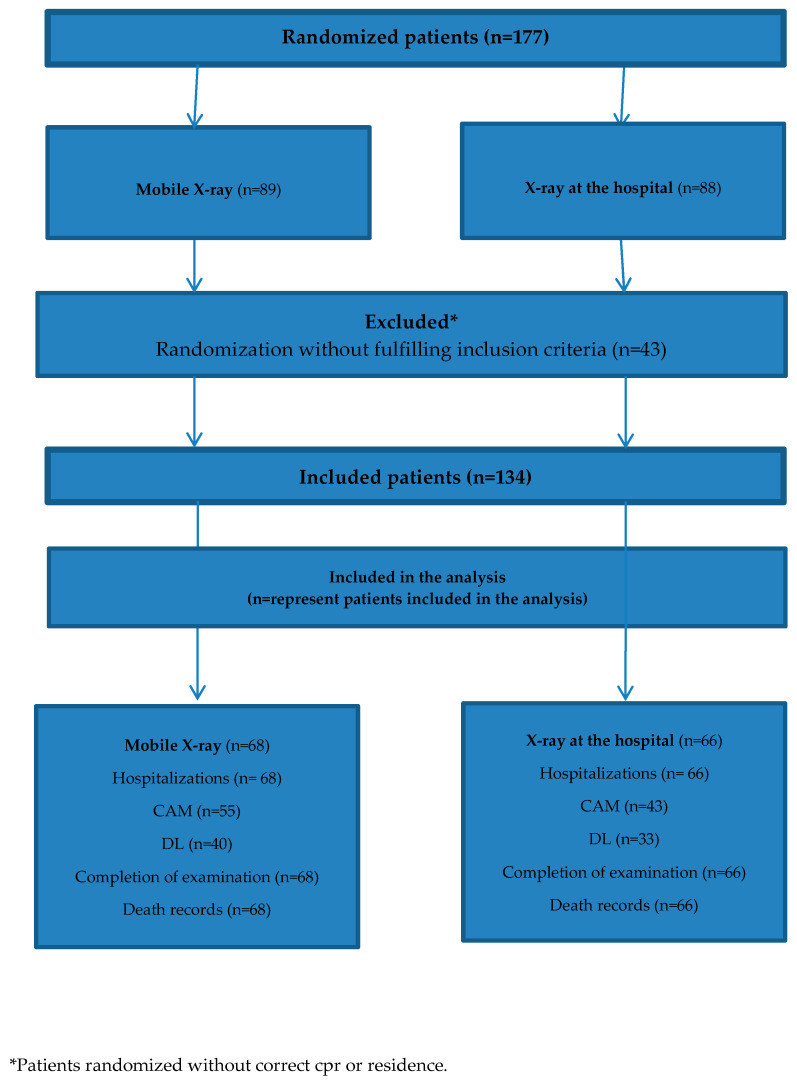
Patient inclusion flow.

**Table 1 healthcare-08-00118-t001:** Baseline characteristics collected interviewing healthcare staff and supplemented with data from local systems.

Characteristic	Group	
	Intervention (n = 68)	Control (n = 66)
Age (mean ± SD)	81.98 (7.28)	84.39 (9.56)
Gender (male)	24 (35%)	24 (36%)
Number of diagnoses (mean)	4.40	4.54
Polypharmacy ≥ 5	48 (71%)	49 (74%)
Patients examined by doctor before X-ray	60 (90%)	56 (85%)
Diagnosed with dementia (yes/no)	27 (40%)	30 (45%)
BMI ≤ 21	40 (59%)	34 (52%)

**Table 2 healthcare-08-00118-t002:** Patient characteristics (mean scores and *p*-values) and impute values for the sensitivity analysis.

Group				
Characteristics n = Included *	Intervention (n = 68)	Control (n = 66)	Scale	Impute Value (Worst Case Scenario in the Dataset)
MMSE *Intervention n = 39 (57%)* *Control n = 30 (45%)*	22.64 (*p* = 0.61)	22.37	0–30	5
GDS *Intervention n = 39 (57%)* *Control n = 30 (45%)*	2.46 (*p* = 0.01)	4.07	0–15 points	14
MBI *Intervention n = 57 (84%)* *Control n = 55 (83%)*	57.52 (*p* = 0.04)	69.88	0–100	5
CIRS G *Intervention n = 57 (84%)* *Control n = 55 (83%)*	11.46 (*p* = 0.51)	12.04	0–30	28

* *p*-value determines if there are statistical significant differences between the intervention group and control group.

**Table 3 healthcare-08-00118-t003:** Number of hospitalizations within one week after the X-ray examination.

X-ray Examination	Hospitalized		
	Yes	No	Total
Mobile X-ray	8 (11.76%)	60	68
X-ray at the hospital	8 (12.12%)	58	66

**Table 4 healthcare-08-00118-t004:** Number of hospital days among the 16 hospitalized patients.

X-ray Examination (n = 16)	Mean	25 Percentile	50 Percentile	75 Percentile	90 Percentile
Mobile X-ray	7.63	3.5	8	12	13
X-ray at the hospital	4.50	1	1	6	19

**Table 5 healthcare-08-00118-t005:** Secondary outcomes measured within one week after X-ray examination.

Outcome Measurement	Intervention Group (n = 68)	Control Group (n = 66)	Scale	Impute Value (Worst Measured Case)
CAM, delirium (yes) *Intervention n = 55 (81%) Control n = 43 (65%)*	0 (*p* = 0.26)	1	Yes or no Yes is worst case	Yes
DL (quality of life), mean *Intervention n = 40 (59%) Control n = 33 (50%)*	7.61 (*p* = 0.05)	5.5	0–30 points 30 is worst case	19
Completion of examination *Intervention n= 68 (100%) Control n = 66 (100%)*	68 (*p* = 0.15)	64	Yes/no	-
Death (%) *Intervention n = 68 (100%) Control n = 66 (100%)*	12 (*p* = 0.59)	10	Yes/no	-

## Data Availability

The dataset used and analysed in the current study is available from the author M.T. on reasonable request and can be found in the computer system Redcap.

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
