# Peer review of "Mobile X-ray Outside the Hospital vs. X-ray at the Hospital Challenges Exposed in an Explorative RCT Study"

_healthcare, 2020, doi:10.3390/healthcare8020118_

Round 1
Reviewer 1 Report
Line 41- Sentences need attention.
Line 51- end of sentence should be changed to ...making their use technically feasible.
Line 83- The study population included...
Lines 86-91- The inclusion and exclusion criteria need to be more precise.
Line 121- So were the data included or not?
Line 166- So what was compared. Were the x-rays match-paired ie. femur/femur?
Lines 167-172- Allowing GPs to influence patient assignment compromises the study. It may be better to eliminate those patients from the study.
Line 177- I could not find what the initials stand for.
Line 181-watch tenses
I agree that design flaws have lessened the value of the study. It also shows that either medical providers were not familiar with the study rules or the rules were not communicated.
Reviewer 2 Report
The article as improved and previous comment were addressed. I just suggest to write as a limitation of the study that the quality of he radiogram was not evaluated in comparison between thew two modalities portable and fixed x-ray apparatus.
Reviewer 3 Report
Overall summary
This manuscript has investigated if mobile X-ray could improve healthcare for fragile patients living in nursing homes and homes for the elderly, compared to X-ray at the hospital. From my point of view, the manuscript has been improved; and, in general, the authors have addressed the changes which were previously suggested. However, as it is noted in detail below, there are a few concerns that should be addressed prior to this paper is considered for publication.
1. Lines 48-51: the references are missing. Please include them. If you include all the references at the end of the paragraph, I could think that they are only for the last sentence but it is not right (see for example, lines 55-57 when you talk about only one cross sectional study).
2. clinicaltrials.gov has not updated the data yet, although I suppose that the information will be updated soon.
3. Lines 136-138. Taking into account the sample size of your study, I cannot understand why the authors include the following sentence: “To show such a difference at a significance level of 5 % (two-sided test) with a power of 80 % would require a sample size of 374 patients (137 patients in each group)(23,2427,28).
4. Please, include the questions used to identify barriers for referring patients (line 162).
5. As you know, the Wilcoxon signed-rank test is used to compare two related samples. Are your samples (ie, control and intervention groups) related or independent? If the samples are independent you must use other non-parametric test.
6. Line 204. SD has not been deleted from Table 5.
7.The changes suggested by the reviewers appear in the text, but without having crossed out/deleted previous sentences/words, so it is quiet difficult to read the article (for example, line 41, the “the” which is before “at the”; line 48, the word “none” which appears before “as”; line 80, the words “patients were” which are after “Aarhus”...). In fact, sometimes I have had to imagine what the authors wanted to say. So, it will be neccessary to pay attention to english language and style when the article is published.

Round 2
Reviewer 1 Report
Check the n= on the Tables.
Improved methods, but not sure how valuable this will be.
Author Response
Dear reviewer,
Thank you for you valuable advice. We have corrected line 150 on p.4. The correct number was 11 and not 12 for MBI and Cirs G. Thank you for notising.
This have been marked with red color.
Sincerely Maria Toppenberg

This manuscript is a resubmission of an earlier submission. The following is a list of the peer review reports and author responses from that submission.
Round 1
Reviewer 1 Report
Lines 62-65 Be more precise. Are you attempting to correlate x-ray location and outcome?
Line 83- Did a BMI of <21 skew the study too much?
Lines 122-124- Considering missing data as worst case scenario may have biased the results. How many missing data cases were involved?
Line 130- Word should be "withdrew"
Line 129-134-Something is missing or out of order here. It does not make sense.
Line 161- By using .05 as the standard for significance, neither of these were statistically significant.
Line 179- "insignificant"
Lines 198-202- Another reason that the results are skewed is that those already at the hospital were easier to keep than those outside the hospital where the easier decision is just to treat in place.
Lines 247-253- These data need further examination. It would help to be precise about withdrawals and those whose physicians pressured researchers to switch groups.
Line 254- Vagueness of the extent of missing variables tarnishes the results.
The paper reads choppy. Review the writing style for better flow.
Reviewer 2 Report
The paper reinforce the clinical impact of X-ray performed in patient environment as an alternative solution to bring patient to the hospital especially to elderly and/or fragile patients.
Definitively bring solution to patient home is more comfortable for patient itself and hospitalization rate after x-ray seems similar in both groups as demonstrate in the paper.
Main lack in the study design is the assessment of the quality of the radiogram. Portable x-ray machine are certainly improved in the recent years but non mobile x-ray apparatus are usually more effective (see reference in attach).
There was a direct reed after performing the examination in patient home sending the images in the hospital for to report to assessment of quality?
Number of repeated radiogram was registered in both solution? please specify.
Minor lack
is unclear in result section why a rib fracture require hospitalization (line 167)

Reviewer 3 Report
This manuscript has investigated if mobile X-ray could improve healthcare for fragile patients compared to X-ray at the hospital. Although this is an interesting study, I feel that:
1. All sections (introduction, methods, results, discussion and even references) need deep changes, so they should be rewritten. For example:
-
In the introduction section: i) the information is not clearly organised, so the story line is not straight; ii) the authors do not talk about advantages/disadvantages of mobile X-ray or X-ray at the hospital (eg. image quality)...
-
In the results section: i) Sometimes, data presented in tables are duplicated in the text or some tables are innecesary (eg. table 4 and lines 164-170); ii) Has the normal distribution of continuous variables been tested? I am not sure, not only because the authors do not explain it, but also because in table 5 both the standard deviation and the percentiles are shown…
-
In general, the discussion section repeats the data which have already been commented in the results section, instead of interpreting them.
2. There are several major flaws in the study premise and design that require attention, and some of which might have led to serious bias in the results. For example:
a) Study population is not defined, and I fear that it is very heterogeneous.
b) Taking into account the challenges (lines 181-190), probably the study design (randomized controlled trial -RCT-) was not the best option. In fact, the authors suggest others study designs (line 245).
c) There are contradictions between the methods described in the manuscript and the methods described in the study registered in clinicaltrials.gov (NCT04005040). For example:
|
|
MANUSCRIPT |
clinicaltrials.gov (NCT04005040) |
|
Study design |
Randomized controlled trial |
Observational |
|
Sample size |
374 |
416 |
d) We do not know what happened with the patients who were randomized by days (lines 96-97).
e) Could the authors guarantee that both groups are similar? I think (see for example, lines 185-187), there are possibilities that the control group has healthier people than the experimental group.
f) There are a large number of missing data, probably resulting in fewer statistically significant results (lines 261-262).
In sum, despite the results of this study (ie, mobile X-ray does not improve healthcare for fragile patients), I would recommend to all fragile patients to carry out mobile X-ray examinations, because from my point of view the results are not entirely reliable.
